# Natural Compounds as Sustainable Additives for Biopolymers

**DOI:** 10.3390/polym12040732

**Published:** 2020-03-25

**Authors:** Nadka Tzankova Dintcheva, Giulia Infurna, Marilena Baiamonte, Francesca D’Anna

**Affiliations:** 1Dipartimento di Ingegneria, Università di Palermo, Viale delle Scienze, Ed. 6, 90128 Palermo, Italy; giulia.infurna@community.unipa.it (G.I.); marilena.baiamonte@unipa.it (M.B.); 2Dipartimento STEBICEF, Sez. Chimica, Università degli Studi di Palermo, Viale delle Scienze, Ed. 17, 90128 Palermo, Italy; francesca.danna@unipa.it

**Keywords:** biopolymers, natural fillers, natural fibers, natural antioxidants

## Abstract

In the last few decades, the interest towards natural compounds, coming from a natural source and biodegradable, for biopolymers is always increasing because of a public request for the formulation of safe, eco-friendly, and sustainable materials. The main classes of natural compounds for biopolymers are: ***(i) naturally occurring fillers (nFil)***, such as nano-/micro- sized layered alumino-silicate: halloysite, bentonite, montmorillonite, hydroxyapatite, calcium carbonate, etc.; ***(ii) naturally occurring fibers (nFib)***, such as wood and vegetable fibers; ***(iii) naturally occurring antioxidant molecules (nAO)***, such as phenols, polyphenols, vitamins, and carotenoids. However, in this short review, the advantages and drawbacks, considering naturally occurring compounds as safe, eco-friendly, and sustainable additives for biopolymers, have been focused and discussed briefly, even taking into account the requests and needs of different application fields.

## 1. Introduction

### 1.1. Increasing Interest towards Natural Compounds

In the last two decades, the extraction and use of naturally occurring compounds experience growing popularity. Interestingly, the keyword ***“Natural Compounds”*** (specifically, keywords in the titles, abstracts, and paper keywords) is contained in about 14,040 scientific publications in 2019, based on Scopus data revelation on January 2020, and, in Figure 1, the trend since 1980 until today of published papers vs. publication years is plotted.

There is evidence that the number of publications increases exponentially in time, especially in the last two decades. Besides, the detected trend, based on Scopus data revelation in January 2020, suggests a continuously increasing interest towards this topic from a scientific point of view. Obviously, numerous scientific papers reveal also properties and performance improvement due to adding of natural compounds to polymers and biopolymers matrices, and this opens new applications for these innovative eco-friendly, sustainable, and safe materials.

However, natural compounds, such as ***natural fillers (nFil), fibers (nFib), and antioxidant stabilizing molecules (nAO)***, are usually added to polymers and biopolymers during their processing and/or manufacturing in order to improve the performance, properties, and durability. Furthermore, in the last two decades, the gradual ***replacement of synthetic polymers with biopolymer counterparts***, i.e., *polymers coming from natural sources*, has gained great interest, considering ever-increasing public interest towards the use of eco-friendly, sustainable, and safe materials, also because of their beneficial effects on human health and environment [1].

### 1.2. Why Add Natural Compounds to Biopolymers?

The biopolymers coming from a natural source are very attractive materials, but overall, as extensively documented in current literature, the biopolymers macroscopical properties and performance, such as rigidity, mechanical and thermo-mechanical resistance, and barrier properties towards gas and water, are limited. Therefore, the drawbacks of biopolymers affect negatively their industrial applications, and it is necessary to add appropriate additives, such as fillers and fibers, in order to gain performance enhancement and large industrial productions [2].

Additionally, the biopolymers have also limited durability in terms of decay of their properties and performance during the service life. Apparently, it seems that durability and biodegradability of biopolymers are contraction terms, but really, there is not; as known, the durability of materials is related to maintaining unchanged properties and performance during materials service life, while the biodegradability of materials is related to their total bioconversion at the end-of-life into renewable resources, such as water, CO_2_, and other natural derivatives. Generally, the maintenance of the biopolymers properties and performance unchanged during the service life can be obtained by adding appropriate stabilizing systems, which are able to protect the organic materials against their oxidative stress [2,3,4,5].

To sum up, the ***naturally occurring*** compounds, such as ***fillers, fibers, and stabilizing molecules***, are added to the biopolymers for three reasons, specifically: *(i)* to formulate bio-based systems, containing only natural compounds, *(ii)* to improve the properties and performance of the biopolymers, in terms of rigidity, resistance, barrier properties, etc., and *(iii)* to keep unchanged the properties and performance of the biopolymers during the service life, ensuring appropriate durability.

### 1.3. Benefits of Adding Natural Compounds to Biopolymers

First of all, the main benefits due to adding of naturally occurring compounds, such as natural fillers, fibers, and stabilizing molecules, to biopolymers are related to the formulation of totally bio-based systems, i.e., at 100% natural matrix and additives, obtaining also complex systems with improved properties, performance, and service life.

However, it is worth noting that the processing and manufacturing of biopolymers through melt processing can be performed profitably using processing technology similar to that of the synthetic polymer-based materials, but unfortunately, as known, the biopolymers *“processing window”* is very strength/short. The fusion temperatures (*T*_m_) and thermal decomposition temperatures (*T*_d_) of biopolymers are very close between them, and, for this reason, their melt processing is a hard matter. For biopolymers, the solvent casting, using different solvents and low environmental temperature, appears to be an easy practicable production way [5]. Moreover, the introduction of appropriate naturally occurring compounds to the biopolymers can be considered a valid practicable way to improve their melt processability, as well as their thermal resistance, properties, and durability, and obviously, the latter leads to enlargement and extension of biopolymers application fields.

The ***natural fillers (nFil)***, such as nano-/micro- sized particles of layered alumino-silicate: halloysite, bentonite and montmorillonite, hydroxyapatite, calcium carbonate, and ***natural fibers (nFib)***, such as wood and vegetable fibers, exert reinforcement and protection actions, increasing significantly the system rigidity, thermal-resistance, and, in some cases, the durability [6,7,8,9]. For example, due to the introduction of fillers and fibers, the elastic modulus and tensile strength of polymer and biopolymer-based systems can significantly enhance, while the reduction of the elongational at break depends significantly on the added amount of fillers and their aspect ratio [7]. According to the literature, the aspect ratio of fillers and fibers is a determining factor for their dispersion and distribution into the organic matrices, and, obviously, the obtained fillers and/or fibers morphology influences significantly the final properties and performance of the polymer and biopolymer-based complex systems, enlarging their application fields.

The natural fillers and fibers are often incompatible with host biopolymer matrices, and to improve their compatibility and dispersion and distribution into an organic medium, two different strategies are usually practiced, specifically: *(i)* introduction of compatibilizing agents; for example, modified polymers containing polar groups [10], and *(ii)* chemical surface treatment; for example, introducing fatty acids onto spherical calcium carbonate (CaCO_3_) particles [11,12], ammonium quaternary salts into alumino-silicates layers [13,14,15], acetylating wood/vegetable fibers [16,17], etc.

Additionally, the presence of natural fillers and/or fibers can significantly limit the humidity and gas penetration into the biopolymers’ structure and bulk, improving their barrier properties and making these materials suitable for some packaging applications [9].

However, the biopolymers, as well as polymers, are subjected continuously to oxidative stress during both processing and service life, as known. The combined actions of atmospheric oxygen, humidity, mechanical stress, heat, and UV-light cause premature loss of the biopolymers’ performance and properties. Really, the chemical composition of biopolymers, as well as of polymers, changes upon oxidative stress, and there is the formation of radicals, unsaturations, oxygen-containing groups, and/or other new chemical groups. Farther, the formation and accumulation of new chemical species lead to the gradual and irreversible changes also at macroscopical level, i.e., the organic materials lose their ductility and become even more rigid, until their total embrittlement. Obviously, to avoid unexpected changes of materials performance and properties, the degradation processes must be monitored and controlled through the introduction of appropriate stabilizing molecules, such as antioxidants and UV absorbers, who are able to stop or at least slow down the formation of radicals, unsaturations, oxygen-containing groups, and new chemical groups [18,19,20].

***Natural antioxidants (nAO)***, such as phenols, polyphenols, vitamins, and carotenoids, are suitable stabilizing systems, which can protect the biopolymers against the thermo- and photo-oxidative degradation, which occurs during manufacturing and in the service life [21,22,23]. As well known, the natural antioxidant compounds are used as suitable additives in the food sector to prevent quick decay of foods and beverages and to extend their storage, and in the pharmaceutical sector to scavenge free radicals and to prevent the tissue lipid oxidation. Some natural antioxidants have been considered as suitable stabilizing systems, also for polymers and biopolymers, in order to prevent their oxidative degradation [23]. Unexpectedly, some naturally occurring compounds, based on their chemical compositions, can cause the faster formation of oxygen-containing groups in the polymer and biopolymer-based systems, inducing premature loss of their performance and properties [23,24,25,26]. Additionally, if the natural fillers (e.g. montmorillonite) contain iron impurities, upon UV-light, the iron ions can change from Fe^2+^ to Fe^3+^, catalyzing the decomposition of the hydroperoxides, which are typical oxygen-containing products formed during the degradation of polymers and biopolymers. Besides, this reaction is auto-catalytic, leading to a significant acceleration of the degradation and loss of the performance for all systems containing this filler kind [27,28].

Overall, this brief review provides an overview of the ***advantages and disadvantages***, coming from the introduction of naturally occurring compounds as sustainable and suitable additives for the formulation of totally biopolymer-based systems. At the same time, it gives notice about some unexpected drawbacks due to the adding of the natural additives into biopolymer matrices, pointing out how the drawbacks can affect negatively the large-scale application of these materials.

## 2. Natural Fillers *(nFil)*

In the past, the introduction of micro-sized particles, such as calcium carbonate, approximately up to 40–50 wt.% to the polymer matrices, aiming to reduce the cost and to increase the rigidity of the systems, has been considered a consolidated practice at large industrial level. Therefore, in the last few decades, the formulation of complex polymer and biopolymer-based systems containing nano-sized fillers has gained great interest because by adding a low amount of nano-fillers, approximately maximum up to 10 wt.%, there is the possibility to obtain properties and performance enhancement, similar to that obtained using micro-fillers at a large amount.

Overall, the adding of micro- or nano-fillers to polymers and biopolymers leads to a significant increase of their rigidity, thermal-resistance, stability, barrier properties, and durability [29,30,31,32,33,34,35,36,37,38,39,40,41,42,43,44,45,46], although in some cases, the presence of the fillers can induce a reduction of some properties due to the development of inappropriate morphology. Besides, the introduction of ***naturally occurring fillers (nFil), such as nano-sized layered alumino-silicates (e.g., halloysite, bentonite, beidellite, sepiolite, montmorillonite, etc.),**hydroxyapatite, calcium carbonate***, etc., to biopolymers offers the possibility to formulate eco-friendly and sustainable materials, also considering the growing public interest towards the formulation of safe materials.

### 2.1. Natural Alumino-Silicates

The alumino-silicates are minerals most abundant in nature and, due to their good properties and low costs, are considered excellent candidates for the formulation of advanced polymer and biopolymer-based nanocomposites for some industrial applications (e.g., automotive components), packaging applications (e.g., food packaging with selective permeability), some outdoor applications (e.g., mulch films), etc. [29,30,31,32,33,34,35,36].

Indeed, to gain a significant increase in the properties and performance of the biopolymers, it is necessary to obtain a good nanofillers dispersion and distribution into the matrices. For example, in the case of layered alumino-silicates, see Figure 2a [32], the achievement of well intercalated and/or exfoliated nanofiller morphology is imperatively required, see Figure 2b [34]. As noticeable, the interface between the nanofillers and polymeric matrix increases significantly due to nanofiller exfoliation, and, subsequently, the nanocomposite mechanical and barrier properties appreciably raise.

Further, to improve the nanofillers dispersion and exfoliation into the polymer and biopolymer matrices, an organo-modification of the layered silicates, through the introduction of ammonium quaternary salts containing long alkyl chains, is usually practiced. The presence of the long alkyl chains between the silicate layers facilitates and helps the layers’ separation during the melt processing and/or manufacturing of the polymers and biopolymers-based nanocomposites, and, obviously, the latter has a great beneficial effect on the complex system properties and performance [29,30,31,32,33,34,35,36].

However, let’s discuss some interesting cases and experimental results documented in the current literature. As reported by Fukushima et al., the adding of natural fillers, such as two organo-modified montmorillonites: Cloisite 30B (CL) and Nanofil 804 (NF), and one unmodified sepiolite (SEPS9) to biodegradable polylactic acid (PLA) and ε-caprolactone (PCL). The introduction of all these kinds of nanofillers leads to an increase of both storage modulus (E’) and tan δ values at low and high temperatures, see Figure 3a,b and Table 1 and Table 2, highlighting the formulation of more stable and thermally resistant nanocomposite systems, in comparison to the neat matrices [37].

Moreover, based on the morphological studies conducted by X-ray diffraction, the authors have reported that the highest thermal-mechanical enhancements have been obtained for PLA-based nanocomposites in comparison to the PCL-based nanocomposites, probably because of higher polymer/filler interactions.

Interestingly, the adding of organo-modified montmorillonite, such as commercial Cloisite 30B (CL), to fully bio-based blends (polylactic acid/polyamide 11, PLA/PA11), where PLA is the dominant component, allows to design and formulate complex systems with improved high temperature creep resistance, see Figure 4 [38]. As documented by Nuzzo et al., the selective positioning of the silicate nanoparticles in the more polar phase, i.e., PA11, and/or at the PLA/PA11 interface turns the blend morphology from droplet to co-continuous at high PLA content, i.e., around 70 wt.%. Particularly, the blend PLA/PA11 = 70/30 wt./wt.% keeps its structural integrity up to 160 °C, taking into account that the neat PLA has a glass transition temperature around 100 °C, see Figure 4a, where the sample deflection as a function of the temperature is shown. Besides, in Figure 4b,c, reported pictures show the samples PLA/PA11 = 70/30 and PLA/PL11 = 70/30 containing silicate at the end of the test, that is after the temperature had reached ≈160 °C.

It is evident that the bio-blend containing silicate show good thermal and dimensional stability at the end of the test at high temperature (approximately at 160 °C), in comparison to the same bio-blend without nanofiller.

Therefore, the preferential location of the nanofillers into the more polar polymer phase and/or at the interface between the two polymer phases, and subsequent morphology variation from droplet to co-continuous, suggests that the nanofillers can be considered as an efficient and suitable physical compatibilizer for some biopolymer blends.

Commonly, the adding of layered alumino-silicates, especially organo-modified layered montmorillonite (OMMT), leads to an increase of the system rigidity, i.e., Young’s modulus (elastic, E), tensile strength (TS), flexural modulus (FM), flexural strength (FS), and, mostly, the ductility of the complex systems, i.e., the elongation at break (EB), remains almost unchanged, see Table 3, reported below [39]. 

Farther, the montmorillonite adding to biopolymers leads to an increase of the barrier properties towards gas and humidity because into the polymeric matrix, the way of the oxygen and water molecules becomes more tortuous. Additionally, the complex system crystallinity could increase if the nFil exerts a nucleating effect, or it could decrease if the nFil prevents and/or inhibits the macromolecules organization in regular crystalline structures [39].

An interesting study by Zhou et al. reported about the increase of the mechanical properties of biopolymer-based foam blends containing polylactic acid, PLA, and polybutylene succinate, PBS. In Figure 5a–c, the compressive strength, bending strength, and impact toughness of biopolymer foam blends-based on PLA/PBS, as a function of the added concentration (wt.%) of natural unmodified montmorillonite (Na-MMT) and organo-modified montmorillonite (OMMT) are plotted. It can be observed that the adding of both Na-MMT and OMMT leads to an increase of the values of the considered compressive strength, bending strength, and impact toughness, in comparison to neat biopolymer-based foam PLA/PBS. As known, the compressive strength, bending strength, and impact toughness are the main critical mechanical parameters for the porous polymer-based materials. Specifically, the compressive strength increases by about 34% and 48% by adding of 3wt.% of Na-MMT and OMMT, respectively, see Figure 5a. Similar increases have been observed also for the bending strength and impact toughness upon the adding of unmodified and organo-modified montmorillonite, see Figure 5b,c, respectively.

Although the adding of nFil leads to a significant increase in the mechanical and thermo-mechanical and mechanical resistance and rigidity of the polymers and biopolymers due to their excellent reinforcement action, i.e., most important beneficial effect, unfortunately, if the nFil contains iron ions, the system durability could be significantly compromised. As known, the iron ions can change from Fe^2+^ to Fe^3+^ upon UV-light, and the latter can catalyze the matrix degradation, limiting in this way, the nanocomposite outdoor applications [27,28]. Additionally, the reduced photo-oxidative resistance of the polymers and biopolymers-based nanocomposites, containing organo-modified alunimo-silicates can be understood considering two different issues: *(i)* presence of iron impurities in the native silicates structures, which are able to catalyze the degradation of the host matrices, and *(ii)* decomposition of the organo-modifier (i.e., surfactant agent) at high processing temperatures, typical for the processing of polymers and biopolymers. Besides, as known, the unwanted phenomenon, related to the decomposition of the organo-modifier (e.g., ammonium quaternary salts decompose following Hoffman’s elimination reaction at high processing temperatures), leads to the formation of α-olefins (a vinyl moiety) and amines, which are able to catalyze further the degradation of the host matrices.

For example, in Figure 6a,b, the trends of dimensionless elongation at break (EB) and Young’s modulus (elastic modulus, E) for neat polyethylene (PE) film and PE films containing unmodified nanofiller (NF), native montmorillonite (MMT), and organo-modified montmorillonite (OMMT), without and with compatibilizer agent, such as maleic anhydride grafted polyethylene (PEgMA), as a function of accelerated UVB-light exposure (using UVB-lamps, having an emission peak at ~313 nm), are plotted. The dimensionless values of EB have been calculated as a value at a given exposure time, EB_(t)_, divided by the value before exposure, EB_(t0)_; the same calculations have been performed also for the trends of elastic modulus. It can be observed that the presence of NF and MMT, even more, the presence of OMMT, leads to a more pronounced decrease of dimensionless EB and more pronounced increase of dimensionless E, highlighting that the presence of nanofillers accelerates significantly the photo-oxidative degradation, i.e., the ductility loss and rigidity increase. Besides, it is evident that these effects are exacerbated by the presence of the compatibilizing agent, such as PEgMA [28].

Further, by Dintcheva et al., it has been demonstrated that the iron ions, if present in the alunimo-silicates structure, see Figure 6c and Table 4, upon UV-light exposure, are able to catalyze the decomposition of the hydroperoxides, which are the typical oxygen-containing products coming from the degradation of the polymeric materials. 

The decomposition of hydroperoxides is significantly exacerbated by the PEgMA presence because it enlarges the nanofiller exfoliation and dispersion, and the latter increases the interface area between the silicate layers and polymeric matrix, which is a critical point for the beginning of the degradation process of the nanocomposites. However, the accelerated photo-degradation of some polymeric nanocomposites, due to the presence of nanofillers, limits significantly their outdoor applications [28].

Therefore, biopolymer-based nanocomposites, containing organo-modified montmorillonite, similarly to the petroleum-based polyolefins upon UV-light exposure, degrade faster than the unfilled matrices because of the presence of some impurities in the structure of alumino-silicates, larger interface surface between matrices and exfoliated nanofillers, and decomposition of the surfactant in the structure of organo-modified alumino-silicates.

An interesting study, related to water permeation through poly(lactic acid)-based nanocomposites containing montmorillonite, has been conducted by Duan et al. [41]. Really, the authors examined accurately how the nanofiller concentration and dispersion, i.e., nano clay morphology, affect the barrier properties and thermal properties of PLA.

Therefore, it is generally accepted that the presence of nanofillers platelets increases significantly the barrier properties of biopolymer-based nanocomposites, in comparison to that of neat matrices. The so-called “tortuous path” model, see Figure 7a, suggests that the rectangular nanoplatelets, aligned perpendicularly to the vapor and/or gases diffusion direction, can make difficult molecules pathway and diffusion into the host matrices. Besides, considering the Nielsen’s equation, reported below, the permeability of polymers and biopolymers composites (P_c_) can be calculated taking into account the value of permeability of neat matrices (P_m_), the volume fraction of fillers (V_f_), and filler aspect ratio (L/D; where L is length, D is diameter/thickness).
(1)$$\frac{P_{c}}{P_{m}}=\frac{1-V_{f}}{1+\left(L/2D\right)V_{f}}$$

Duan et al. [41] reported experimental proof, obtained by Transmission Electron Microscopy (TEM), about the obtained alignment of nanofiller particles into PLA-based nanocomposites, achieved following melt processing and compression molding, see Figure 7b.

The water vapor rate transmission (WVRT), i.e., water permeability, of PLA-based nanocomposites decreases due to the presence of nanofillers particles, and this decrease is more pronounced for samples containing large number of nanoparticles, see Figure 7c. As expected, the water vapor barrier of PLA-based nanocomposites, being the reverse of WVRT, significantly increases due to the presence of aligned nanofillers platelets.

However, similarly to the water vapor molecules, also the gas molecules have a tortuous pathway into the nanocomposites, rather than into the neat matrices. The biopolymer-based nanocomposites can be considered suitable and valid candidates for different packaging applications, due to their enhanced barrier properties, in comparison to neat matrices.

### 2.2. Natural Hydroxyapatite

The poly-L-lactic acid (PLLA) is a semi-crystalline thermoplastic, having excellent biocompatibility and biodegradability. This biodegradable polymer is suitable for different biomedical applications, such as porous biodegradable scaffolds for bone repair, where the rate of resorption in vivo vs. growth rate of the bone tissue must be imperatively monitored. The adding of hydroxyapatite (nHA), as suitable, sustainable, non-toxic, and bio-active nanofiller, has beneficial effects on the thermo-mechanical, mechanical, and resorption behavior of the PLLA-based nanocomposites for biomedical applications, such as porous biodegradable scaffolds for bone repair.

An interesting study by Delabarde et al. [42] reported that the presence of nHA in PLLA, although leads to a slight increase of the system rigidity and maintenance of the ductility, see Figure 8a,b, it leads to an acceleration in the mass loss, and this effect is more pronounced for amorphous PLLA samples, see Figure 8c. Besides, the degree of crystallization of neat amorphous PLLA increases with increasing the aging time, while the adding of nHA leads to a decrease of the crystallinity of the PLLA, and this decrease becomes more pronounced with increasing the aging time, see Figure 8d. As discussed above, the well-dispersed nHA inhibits the organization of the macromolecules, and, subsequently, the nanofiller samples show a lower degree of crystallinity, in comparison to the neat PLLA one, and, besides, this effect becomes more pronounced at longer aging times. 

Finally, the authors concluded that the addition of nHA results in significantly less marked decreases in tensile strength and strain to failure as a function of aging time than the unmodified films. This unexpected result could be understood considering that the adding of nHA exerts a beneficial effect on both the toughness and ductility of PLLA-based film. The latter is of considerable interest, given that a primary aim in bioresorbable scaffolds is to maintain structural integrity at intermediate stages of resorption.

Currently published study reports about the improvement of mechanical, thermal, and combustion resistance of polylactic acid (PLA) through adding of hydroxyapatite (HA) nanoparticles, previously modified phosphorus-based organic additive (PDA) [43]. Due to the adding of organo-modified HA nanoparticles, the main mechanical properties, such as tensile strength (TS) and Young’s modulus (YM), of nanocomposites increase, while, the ductility, i.e., the elongation at break remains almost unchanged, see stress-strain curves in Figure 9a,b and main mechanical properties in Table 5. 

Interestingly, also according to other studies reported in the literature, the presence of untreated HA nanoparticles increases the thermal and combustion resistance of PLA-based nanocomposites, while the presence of phosphorus-based organic modified HA advances slightly the thermal and combustion degradation of PLA because of the presence of the organo-modifier. The latter effect is very similar to that observed for the organo-modified alumino-silicates, and, unfortunately, this could compromise the applications of the nanocomposites in some specific fields, where an appropriate resistance to oxidation and/or degradation phenomena is required.

### 2.3. Natural Calcium Carbonate

The nanoparticle morphology, as mentioned before, is a determining factor for the achievement of enhanced properties and performance of the biopolymer-based nanocomposites, in comparison to the neat matrices. To improve the nanoparticles dispersion and distribution and to prevent the potential re-aggregation during melt processing, the nanofillers are often organo-modified, and, specifically, in the case of calcium carbonate, usually considered industrial practice is the introduction of fatty acids onto surface of the spherical particles. Really, the presence of fatty acids helps the dispersion of the calcium carbonate particles into the polyolefins in an efficient way, while, in the case of biopolymer matrices, the fatty acids can react with some specific functional groups, penalizing few macroscopical properties. Therefore, the application fields of calcium carbonate-containing biopolymer-based nanocomposites are very promising from industrial applications in the transport and construction sectors to the packaging and agricultural sectors.

Nekhamanurak et al. investigated the effect of the presence of micro- and nano-sized CaCO_3_ on thermal stability and melt rheology behavior of PLA [44]. Surprisingly, the authors found that the micro- and nano- composites show lower thermal stability, i.e., the onset values of the thermal degradation of the composites are lower than the value of neat PLA, see Figure 10. As noticeable in Figure 10, the decomposition temperature (*T*_d_) for neat PLA is around 392 °C, while the *T*_d_ for the micro- and nano-composites are around 295–356 °C, highlighting a significant reduction. Unfortunately, the chemical reaction between fatty acid and ester linkage of PLA induces chain scission and causes thermal degradation of PLA.

Another similar study by Kumar et al. [45] reported that the degradation of PLA-based nanocomposites occurs at lower temperatures than that of neat PLA one. In Figure 11 and Table 6, the TGA curves and temperatures at which the samples’ weight is reduced at 10% and 50% of their initial values are reported. As noticeable, the reduction of 10% of sample weight for PLA-based nanocomposites occurs at lower temperatures for the nanocomposites containing calcium carbonate (*T*_10_) compared to the unfilled PLA one. This could be understood considering that the degradation process of the organo-modifier of the nanoparticles begins before the degradation of the PLA.

Besides, the nanocomposites are more susceptible to oxidation also because the presence of the nanoparticles hinders the organization of the macromolecules in regular structures, and, overall, the crystallinity degree decreases. Interestingly, the reduction of the sample weight at half (i.e., T_50_) for neat PLA and PLA-based nanocomposites occurs at very close temperatures because, at higher temperatures, the influence of the presented fatty acids onto calcium carbonate particles is significantly reduced. Upon further temperature increase, as expected, the reduction of sample weight is more pronounced for neat PLA.

Therefore, the effect of the micro- and nano-sized calcium carbonate on the thermal stability of the composites can be explained considering two complementary effects: *first*, the CaCO_3_ particles are typically treated by fatty acid to prevent their re-agglomeration during storage and handling, and *second*, the presence of both untreated CaCO_3_ or organo-modified particles can decrease the crystallinity degree of the host matrices. The possible interaction/reaction between the fatty acids and host polymer matrices depends significantly on the chemical nature of presented functional groups onto the polymer structures, for example, the petroleum-based polyolefins, such as polyethylene, polypropylene, polystyrene, do not react with the fatty acids, while the bio-polyesters, having numerous esters groups, are able to react with the fatty acids, inducing a negative effect due to the adding of calcium carbonate particles. Additionally, the presence of particles into semi-crystalline host polymer matrices could prevent the organization of the macromolecules in ordered crystalline structures, reducing in this way the composite crystalline degree. As known, the amorphous structures are subjected to faster oxidation, which occurs also at lower temperatures, in comparison to neat matrices, because of easy oxygen penetration, highlighting the formation of less thermal-resistant structures.

### 2.4. Nanocrystalline Cellulose

The nanocrystalline cellulose (NCCs) is generally named rigid rod-like nanoparticles having diameters 10–20 nm and length of a few hundred nanometers [46]. As known, the NCCs contain numerous hydroxyl groups, which can be subjected to different chemical surface treatments. The production of NCCs includes two different preparation techniques: first, the removal of amorphous portions between the natural fibers’ structure by acid treatments, and, second, bacterial synthesis. The bacterial cellulose is a highly pure form, i.e., free of hemicellulose, lignin, and pectin, and, additionally, the bacterial cellulose membranes are highly porous materials, having high liquids and gases permeability and high-water uptake.

According to the literature, to produce advanced nanocomposites, the NCCs are successfully introduced in polylactic acid (PLA), polycaprolactone (PCL), and thermoplastic starch (TPS) as suitable reinforcement agents. Therefore, the final performance of NCCs-containing polymers and biopolymers nanocomposites depends on nanoparticle aspect ratio, surface area, and its dispersion in the matrix. The retaining of nano-dimensions, i.e., as separate nanoparticles and uniform dispersion in polar and nonpolar matrices, is a challenge, and, as currently reported in the literature, different preparation techniques have been adopted to overcome the nanoparticles’ aggregation.

Therefore, the applications of NCCs-containing nanocomposites at the industrial scale are still under investigation and must be established and exploited, although there have many promising achievements at the laboratory or pilot scale. Interestingly, the NCCs, together with alunimo-silicates or carbon-based nanofillers, are promising additives for biopolymers for the formulation of advanced nanocomposites, having improved properties and performance [46].

Finally, to sum up, the nFil influences the performance and properties of biopolymers in different ways, as follows:*(i)* nanocomposites rigidity increases without ductility loss, i.e., the elastic modulus and tensile strength increase, while the elongation at break remains almost unchanged;*(ii)* barrier properties towards water and gasses molecules increase because the presence of well-dispersed and exfoliated nanoparticles makes the tortuous way of all kind of molecules;*(iii)* thermo-mechanical resistance, deflection temperature, and dimensional stability significantly increase;*(iv)* thermal and crystallization behavior could be accurately evaluated; the nFil can exert nucleating effect (enhancing the crystallinity) or can hinder the organization of the macromolecules in the crystalline structures (decreasing the crystallinity);*(v)* durability and resistance to oxidative stress can be penalized if the nFil is organo-modified or, in its composition, is present some impurities and/or contaminants.

## 3. Naturally Occurring Fibers (nFib)

The development of high-performance materials based on natural resources is a world-widely increasing issue, as mentioned above. Different ***natural fibers (nFib)***, **i.e., *wood and vegetable fibers, such**as**flax, hemp, jute, kenaf, sisal, bamboo, etc.***, are widely taken in consideration for the formulation of eco-friendly, sustainable, and safe biopolymer-based materials [47,48,49,50,51,52,53,54,55,56,57,58,59,60,61,62,63,64,65,66,67,68,69,70,71]. Interestingly, nowadays, ***abaca, pineapple leaf, coir, oil palm, bagasse, and rice husk fibers*** have also gained interest and importance because of their availability and specific properties. Therefore, the biopolymers containing naturally occurring fibers have numerous applications in different fields, such as automotive, constructions and indoor decors, packaging with selective permeability, etc.

It should be mentioned that the nFib is a multi-component system and is mainly composed of cellulose, hemicellulose, lignin, and waxes, in different weight percentage, as reported in Table 7 [71]. Therefore, the nFib shows highly variable properties, such as sensitivity to temperature, moisture, and UV radiation, because their location and time of harvest are totally different. Besides, due to different nFib compositions, the nFib is not fully compatible with host polymers and biopolymer matrices, and, for this reason, an appropriate physical and/or chemical surface treatment of the fibers and/or use of compatibilizers between fibers and matrices are usually practiced [47,48,49,50,51,52,53,54,55,56,57,58,59,60,61,62,63,64,65,66,67,68,69,70,71].

Due to variable compositions and properties of different kinds of natural fibers, the effect of the fibers on the properties and performance of the host matrices need to be evaluated for each individual case. For example, the flax or sisal fibers can exert a reinforcement effect on the polylactic acid (PLA) matrix, but, based on their different aspect ratio and compositions, can exert a different effect on the thermal and crystallization behavior. 

An interesting study by Oksman et al. [50] reported about the optimization of the mechanical properties of PLA/flax composites, without and with plasticizer, such as triacetin. Besides, the properties of PLA/flax and PP/flax systems have been compared, and the obtained results suggest that the flax fibers are appropriate reinforcing additive for PLA, like PP.

In Figure 12a–c, the main mechanical properties, such as tensile stress, tensile modulus, and unnotched Charpy impact strength, are reported for neat PLA and PLA/flax composites, containing also different amount of plasticizer, such as triacetin (0–15 wt.%). As known, PLA is a brittle biopolymer, and to improve its processability and ductility, the suitable plasticizer is usually added. Based on experience and obtained results, the authors assert that the efficient amounts of flax fibers and plasticizer, in terms of mechanical performance enhancement, are at 30–40 wt.% and 12–15 wt.%, respectively. As noticeable in Figure 12a, the tensile stress decreases with increased triacetin content, and this trend is even more pronounced in PLA/flax composites, rather than in neat PLA. Similar considerations can be made, also for the stiffness trends, and the triacetin presence affects negatively the stiffness of PLA/flax composites, while it does not affect the stiffness of neat PLA, see Figure 12b. The impact values of PLA/flax remain almost unchanged due to the presence of triacetin, while a slightly pronounced beneficial effect of plasticizer is noticed for neat PLA, see Figure 12c.

A very interesting study by Graupner et al. [67] deals with the design and tunes the properties of PLA-based composites containing different kind of fibers. The authors assert that by learning the role and function in plants, there is a possibility to design biopolymer-based composites by adding seed fibers with high elongation for improved impact and stem fibers for improved stiffness. In Figure 13a–d, the tensile strength, Young’s modulus, elongation at break, and Charpy impact strength for PLA-based composites containing different kinds of fibers are reported. The dashed lines represent the main mechanical characteristics of neat PLA, while, due to different fibers orientation, for all the PLA-based composites, there is the difference between the values in the machine direction (MD) and cross direction (CD).

As noticeable in Figure 13, the best characteristic values are obtained in Lyocell-PLA composites, although no good bonding between the matrix and fibers occurs. Cotton-PLA composites show high elongation at break, while the hemp, hemp/kenaf, and kenaf-PLA composites show good results in terms of tensile strength, elongation at break, and elastic modulus values, in comparison to cotton/PLA one. The kenaf fibers—added as single fiber or together with other fibers—could act as a crack inhibitor, at the same time, causing an impact improvement for PLA. Unfortunately, the authors assert that by modeling the tensile strength of the composites through the rule of mixture, the calculated values are multiple times higher than the measured values. A second important point of this research is related to the adding of a combination of two different kinds of fibers; for example, the best results in terms of tensile properties are obtained by adding hemp and Lyocell to PLA.

Therefore, to improve the compatibility between the nFib and host polymer or biopolymer matrices, usually, specific compatibilizing agents, such as adhesion promoters, coupling agents, or plasticizers, are introduced during the composites processing [71]. Additionally, the chemical treatment of the natural fibers is usually practiced, although it is not enough for good compatibilization and efficient force transmission. The chemical treatment of the natural fibers, usually, is a good practice to eliminate some contaminants of the fiber surface and to create new chemical groups that are more compatible with the groups presented onto the macromolòecuels or onto the compatilizing agents.

Finally, to sum up, the adding of nFil to biopolymers is a very attractive field, having positive effects on the system rigidity and impact properties (crack inhibition), the system elasticity, i.e., the elongation at break, if the composites are significantly reduced. The addition of appropriate adhesion promoters, coupling agents, or plasticizers can further improve the stress force transmission of composites, amplifying the composite application fields.

## 4. Naturally Occurring Antioxidant Molecules (nAO)

Since ancient times, the natural antioxidant molecules have been extracted by plants, vegetables, and fruits and used because of their excellent health effect, such as anti-aging, anti-bacterial, antioxidants, etc. Farther, due to the powerful antioxidant effect of numerous naturally occurring molecules, currently, they are considered as stabilizing molecules for food and beverage storage and as efficient anti-cancer agents [72,73]. Therefore, in the last two decades, the nAO is also proposed and used as stabilizing molecules for polymers and biopolymers because of their efficient antioxidation and protection abilities, similar to those exerted by the synthetic stabilizing molecules [74,75,76,77,78,79,80,81,82,83,84,85,86,87,88,89,90,91,92,93,94,95,96,97]. The main classes of nAO considered for the protection of biopolymers are the natural phenols and polyphenols, vitamins, and carotenoids. The application fields of biopolymers containing nAO are numerous, for example, safe food packaging, mulch films, pharmaceuticals and cosmesis, biomedical devices, human body tissue, and implants, etc.

However, phenols and polyphenols exist in a large variety in nature; they account for approximately 200,000 [23]. Really, their classification depends on the number of phenolic units: ***phenolic acids*** contain one phenol functionality; ***flavonoids and stilbenes*** contain two phenol subunits; ***tannin*** and other – more phenol subunits, see Table 8. The ***phenolic acids*** are classified into two main groups: ***hydroxycinnamic acids****, such as p-coumaric acid, caffeic acid, ferulic acid, sinapic acid, etc., and **hydroxybenzoic acids**, such as p-hydroxybenzoic acid, vanillic acid, gallic acid, syringic acid,* etc., see Table 8.

***Vitamins,** such as Vitamin A, C, and E*, and ***carotenoids**, such as β-carotene, lycopene, lutein, etc.*, are very popular and used in food, pharmaceutical, and cosmetic industries as protective agents, colorants, and food supplementary. Currently, vitamin E and β-carotene, chemical formulas shown in Table 3, are successfully used as melt processing stabilizers and as UV-light protector, respectively, for both polymers and biopolymers.

However, the nAO is able to scavenge and react with the free radicals and oxygen- and nitrogen-containing species. The nAO can act as chain-breaking antioxidants, i.e., named primary AO, or as protective antioxidants, i.e., named secondary AO, and, really, numerous of them have multiple functions because their protective actions are exerted by both primary and secondary protective ways [74,75,76,77,78,79,80,81,82,83,84,85,86,87,88,89,90,91,92,93,94,95,96,97].

In the current scientific literature, different studies about the stabilization of the main classes of biopolymers having an industrial impact, i.e., starch-based biodegradable polymers and biopolyesters, using naturally occurring molecules, are published [74,75,76,77,78,79,80,81,82,83,84,85,86,87,88,89,90,91,92,93,94,95,96,97]. All these papers agree that the natural stabilizers are able to protect the biopolymers against the oxidative stress in a similar way as the synthetic antioxidant and/or light stabilizers.

In particular, the starch-based biodegradable polymer, named Mater-Bi^®^ (MB), has been added with the extract of polyphenols and with quercetin (Q) and vitamin e (V-E) [79], in order to protect the MB against both thermo-mechanical-oxidation during melt processing and photo-oxidation in service life. Thin neat MB film and MB films containing different nAO and commercial synthetic antioxidants (AO) and light stabilizers (LS) have been subjected to UV-light exposure using UVB-lamps having an emission peak at 313 nm. In Figure 14a,b, the trends of dimensionless elongation at break (EB), i.e., the ratio between the values of the EB_(t)_ at given exposure time and the EB_(t0)_ before exposure, of MB as a function of the irradiation time are shown, and the results are compared with those obtained using synthetic stabilizers. As known, the trends of elongation at break as a function of the exposure time is the most important mechanical parameter for some outdoor applications of the films, especially as mulching and coverage agricultural films.

In Figure 14a,b, it can be observed that the trends of the dimensionless EB in the presence of the natural antioxidants (nAO) and synthetic AO and LS are very similar, highlighting good protection abilities of Q and V-E, comparable to those offered by the synthetic molecules. The latter could be understood considering that the natural antioxidants are able to entrap radicals, formed during melt processing of MB, and protect in an efficient way the MB films, although the best results have been obtained using quercetin.

In another study, to protect efficiently the biopolyester, i.e., polylactic acid (PLA), using nAO, the PLA has been additivated with resveratrol (R) at 0, 1, and 3 wt.% and prepared thin films have been subjected to UV-light exposure [82]. In Figure 15a–c, the trends of carbonyl accumulations, molecular weight variations, and elongation at break (EB) of PLA/R in comparison to neat PLA as a function of the irradiation time are shown. According to the literature, the hydrolytic degradation pathway of PLA, upon environmental humidity, leads to the formation of carboxylic acids end groups, while the main degradation pathway of PLA, upon UV exposure, proceeds with hydrogen abstraction on the polymer backbone and subsequent formation of radicals, which upon reacting with oxygen form peroxy radicals and hydroperoxides. The decomposition of the peroxy radicals and hydroperoxides, through β-scission reactions, leads to the formation of anhydrides. Therefore, the actions of both mentioned mechanisms lead to the significant decrease of PLA molecular weight due to the fragmentation of the macromolecules. Overall, as known, the UV exposure of polymers and biopolymers causes the formation of new chemical groups, e.g., carbonyl and hydroxyl functionalities, and, contemporarily, macromolecular chain scission, which leads to the loss of the macroscopical performance, such as reduction of elongation at break. In Figure 15a,b, the trends of carbonyl accumulation and molecular weight variation of neat PLA and PLA/R are reported, and it can be observed that the presence of R leads to less pronounced accumulation of carbonyl functionalities and reduction of the molecules’ weight, highlighting the protection ability of R towards UV irradiation. Additionally, the EB for neat PLA film goes to very low values at short exposure time than the samples of PLA containing 1 and 3 wt.% of R, suggesting again a good protection ability of R towards irradiation, see Figure 15c. As expected, the protection ability of R is even more pronounced at high R concentration. As mentioned above, the photo-oxidation process of PLA upon UVB exposure proceeds with random chain scission and formation of anhydrides and the formation and decomposition of hydroperoxides, leading to the formation of esters and unsaturations, if PLA is subjected to UVA [93].

Therefore, if the PLA is subjected to UVB exposure, the variation of the anhydride absorption band can be considered a valuable tool to follow the photo-degradation process. In Figure 16, the variations of anhydride band area for PLA sample, added with (a) 0.5 and 1 wt.% [96] and (b) 2 and 3 wt.% of ferulic acid (FA) [26], are plotted.

First of all, there is a need to take into account that the thickness of PLA samples in (a) and (b) are different, and, for this reason, the irradiation times are different. However, in Figure 16a, it can be observed that the adding of 0.5 wt.% of FA is able to protect the PLA, while the adding of 1 wt.% of FA leads to a larger accumulation of anhydrides, and, surprisingly, the FA exerts a concentration-dependent anti-/pro-oxidant behavior [93]. In Figure 16b, it can be clearly observed that doping PLA with 2 and 3 wt.% of FA leads to a large accumulation of anhydride functionalities, and, additionally, no significant difference is noticeable in the trends for PLA containing FA at 2 and 3 wt.% [26]. 

The pro-oxidant activity of naturally occurring antioxidant is a surprising effect for biopolymers, but there is a well-known effect in medicine. Besides, Paracelsus (1493–1541) spoke that the toxicity of naturally occurring compounds is the matter of dose, and the toxicological risks arise if the daily doses are above threshold limits. Additionally, there is a need to take into account that the balance between the benefits and toxicity of natural compounds depends on their oxidative activity, concentration, circumscribed conditions, such as pH, temperature, other active species, etc.

Therefore, in the current literature, are available also other studies that report about the pro-oxidant activity of the naturally occurring antioxidant, for example, vitamin E and β-carotene [23]. All authors agree that at high processing temperatures, typical for the processing of polymers and biopolymers, the nAO can activate H-atoms abstraction from the macromolecules, which induces faster radical formations, and, in the presence of oxygen, the formation of oxygen-containing products, leading to premature pro-oxidation. Really, this mechanism could be considered a valuable tool to accelerate the accumulation of the oxygen-containing groups for the biodegradable polymers, opening a new avenue toward the control of the biopolymer degradation time.

## 5. Conclusions Remarks and Future Perspectives

The interest towards naturally occurring compounds for biopolymers and polymers is always continuous because of a public request for the formulation of safe, eco-friendly, and sustainable materials. As discussed above, the naturally occurring compounds for biopolymers are of three kinds:*(i)* naturally occurring fillers (nFil), such as nano-/micro-sized layered alumino-silicate: halloysite, bentonite, montmorillonite, hydroxyapatite, calcium carbonate, etc.;*(ii)* naturally occurring fibers (nFib), such as wood and vegetable fibers; *(iii)* naturally occurring antioxidant molecules (nAO), such as phenols, polyphenols, vitamins, and carotenoids, highlighting the main advantages and disadvantages in considering them as sustainable and suitable additives.

The presence of nFil and nFib, if they are dispersed and distributed appropriately, can enhance significantly the properties and performance, such as mechanical and thermo-mechanical resistance, barrier properties towards water and gasses, and durability of the formulated composite materials, in comparison to that of the neat host matrices. Unfortunately, if the nFil and nFib contain impurities and/or organo-modifiers (required to improve the compatibility and dispersion with host matrices), the durability of the complex systems could be penalized because of unwanted interactions and/or reactions with some functional groups present in the biopolymer macromolecules.

The nAO can protect efficiently the biopolymers against their thermal- and photo-oxidative stress, during both processing and service life, through the well-known antioxidant mechanism. Unfortunately, if the nAO is added at high concentrations to biopolymers, it can induce faster H-atom extraction from the macromolecules and can exert a pro-oxidant effect. Obviously, controlling the concentrations of nAO and circumscribed conditions, the nAO pro-oxidant effect could be appropriately considered for controlling the degradation times of biopolymers.

Finally, in this short review, the advantages and drawbacks, considering naturally occurring compounds as safe, eco-friendly, and sustainable additives for biopolymers, have been focused and discussed briefly, even considering the requests and needs of different application fields.

The future perspectives, in the opinion of authors, in using naturally occurring compounds as suitable additives for biopolymers is related to three main issues: (i) exploring and discovering new natural additives, (ii) improving bio-based materials’ properties and performance through interface engineering and employing new additives, and (iii) achieving target properties in order to extend the applications fields of bio-based materials containing only naturally occurring additives.

## Figures and Tables

**Figure 1 polymers-12-00732-f001:**
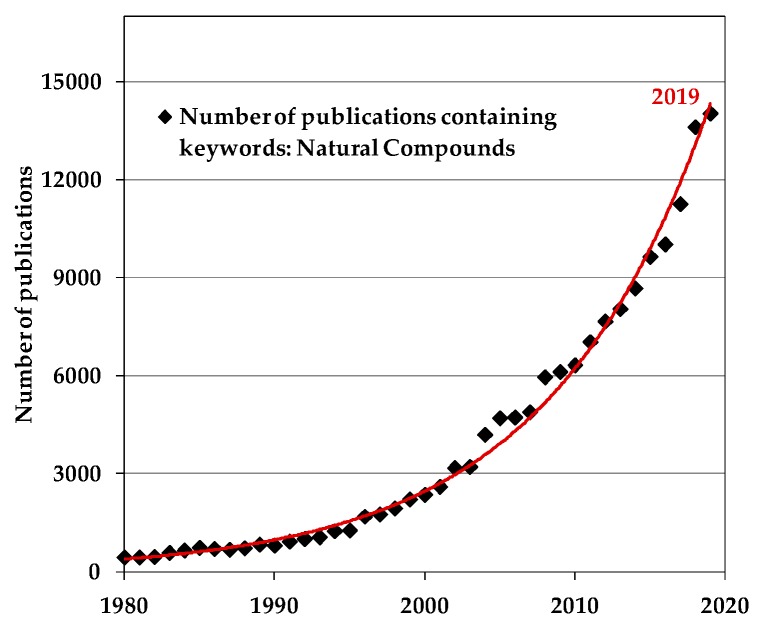
The trend of published papers vs. publication year (Scopus data revelation in January 2020).

**Figure 2 polymers-12-00732-f002:**
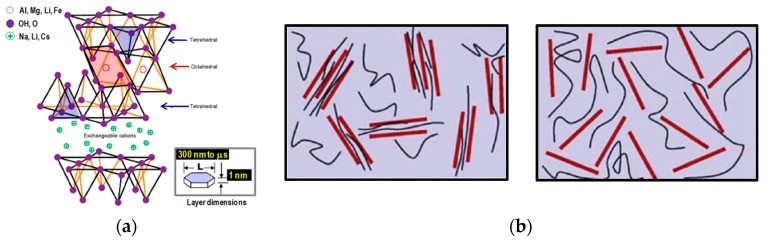
(**a**) Structure of layered alumino-silicate and (**b**) intercalated (left) and exfoliated (right) morphology of layered alumino-silicate into polymer and biopolymer matrices [32], *Reproduced with permission from Bayer G, Plast Addit Compound; published by Elsevier, 2002*.

**Figure 3 polymers-12-00732-f003:**
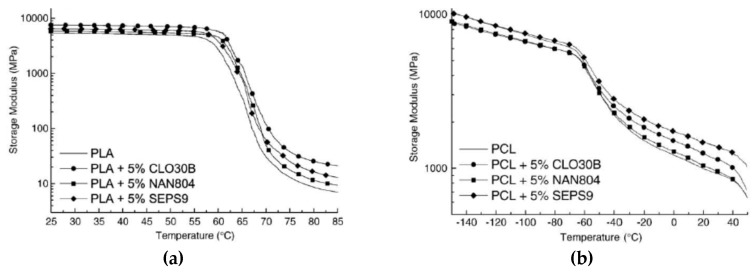
Storage modulus, E’, for (**a**) polylactic acid (PLA) and (**b**) PCL-based nanocomposite systems [37], *Reproduced with permission from Fukushima K, Mater. Sci. Eng. C; published by Elsevier, 2009*.

**Figure 4 polymers-12-00732-f004:**
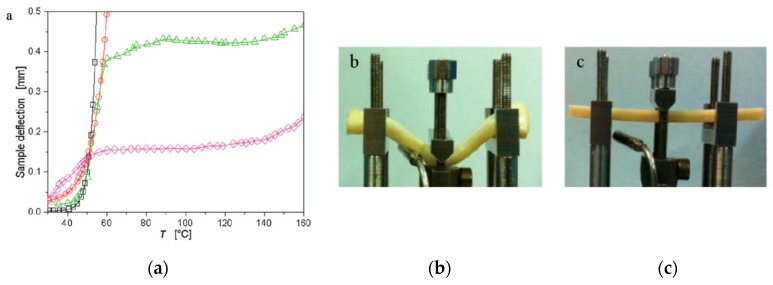
(**a**) Sample deflection recorded during creep tests for the sample PLA (squares), polyamide 11 (PA11) (diamond), PLA70 (circles), and PLA70-silicate at 3 wt.% (triangles). (**b**,**c**) The pictures show the samples PLA70 and PLA70-silicate at the end of the test, that is, after the temperature had reached ≈160 °C [38], *Reproduced with permission from Nuzzo A, Macromol. Mater. Eng.; published by Wiley, 2014*.

**Figure 5 polymers-12-00732-f005:**
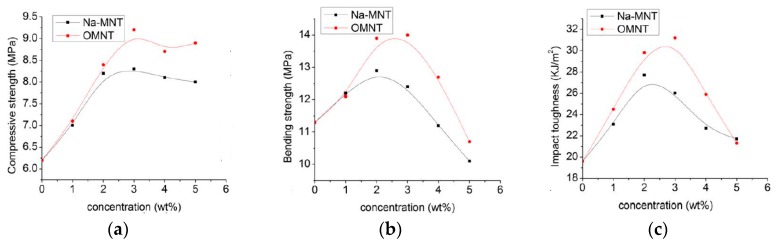
Effect of natural unmodified montmorillonite (Na-MMT) and organo-modified layered montmorillonite (OMMT) on mechanical properties: compressive strength (**a**), bending strength (**b**) and impact toughness (**c**) of PLA/polybutylene succinate (PBS) foams [40], *Reproduced with permission Figure 2014*.

**Figure 6 polymers-12-00732-f006:**
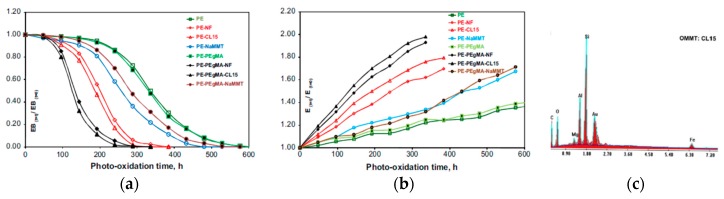
(**a**) Elongation at break (EB) and (**b**) Young’s modulus (E) of different samples without and with montmorillonite and compatilizing agent as a function of accelerated weathering exposure, using UV-B lamps (~313 nm). (**c**) energy dispersive X-ray analysis (EDX) of Cloisite 15A [28], *Reproduced with permission from Dintcheva NTz, Polym. Degrad. Stab.; published by Elsevier, 2009*.

**Figure 7 polymers-12-00732-f007:**
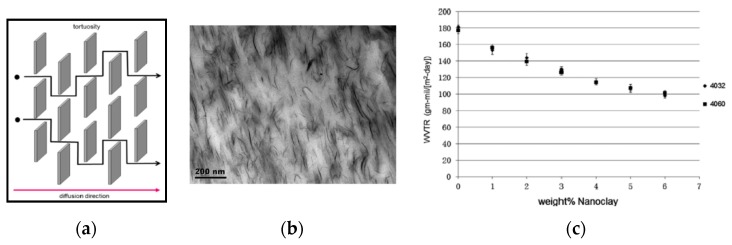
(**a**) Schematic representation of the “tortuous path” model for nanocomposites. (**b**) Transmission electron micrograph (TEM) of PLA nanocomposite containing 3 wt.% of montmorillonite, showing aligned clay platelets. (**c**) Water vapor transmission rates (WVTR) of PLA nanocomposites as a function of wt. nanofillers for two different PLA grades [41], *Reproduced with permission from Duan Z, J. Membrane Sci.; published by Elsevier, 2013*.

**Figure 8 polymers-12-00732-f008:**
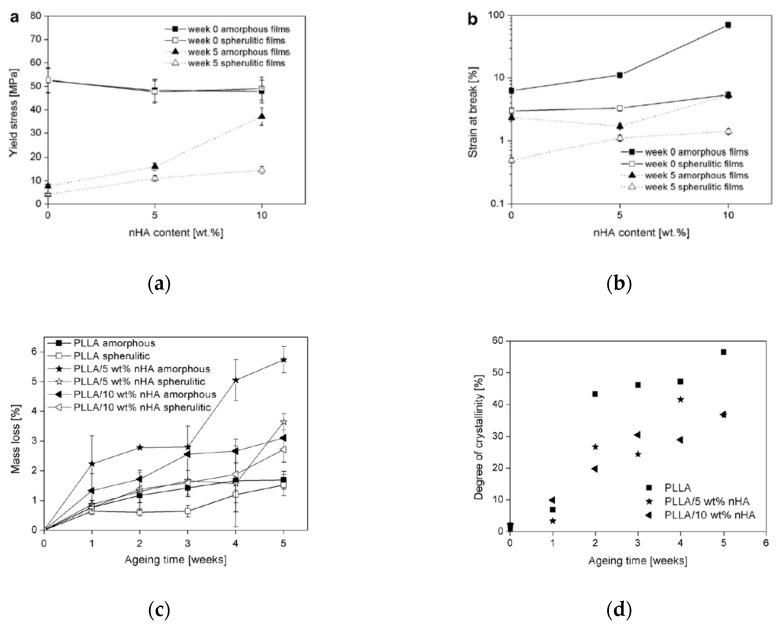
(**a**) Yield strength and (**b**) strain at break for poly-L-lactic acid (PLLA) samples before and after 5 weeks of aging as a function of hydroxyapatite (nHA) content. (**c**) Mass loss and (**d**) degree of crystallinity for amorphous sample, as a function of the aging time for PLLA containing nHA at 0, 5, and 10 wt.% [42], *Reproduced with permission from Delabarde C, Polym. Degrad. Stab.; published by Elsevier, 2011*.

**Figure 9 polymers-12-00732-f009:**
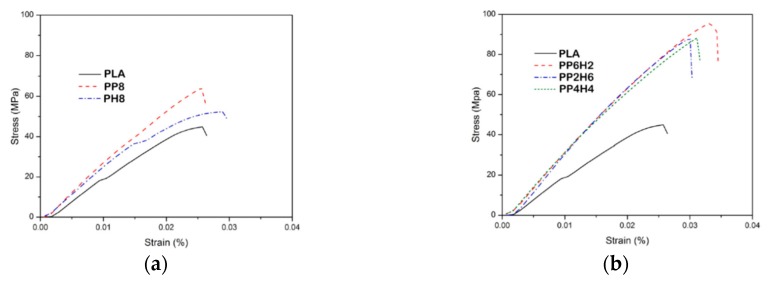
(**a**,**b**) Stress-strain curves of neat PLA and PLA containing hydroxyapatite (HA) nanoparticles modified with phosphorus-based organic additives [43], *Reproduced with permission from Hajibeygi M, Polym. Adv. Technol.; published by Wiley, 2019*.

**Figure 10 polymers-12-00732-f010:**
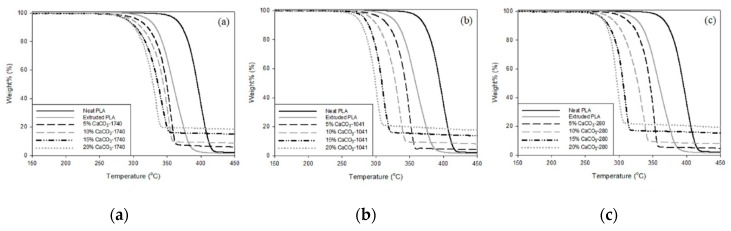
TGA curves of the CaCO_3_-PLA bio-composites with various loading of CaCO_3_: (**a**) micro-CaCO_3_-1740; (**b**) nano-CaCO_3_-1041; (**c**) nano-CaCO_3_-280 [44], *Reproduced with permission from Nekhamanuraka B, Energy Procedia; published by Elsevier, 2014*.

**Figure 11 polymers-12-00732-f011:**
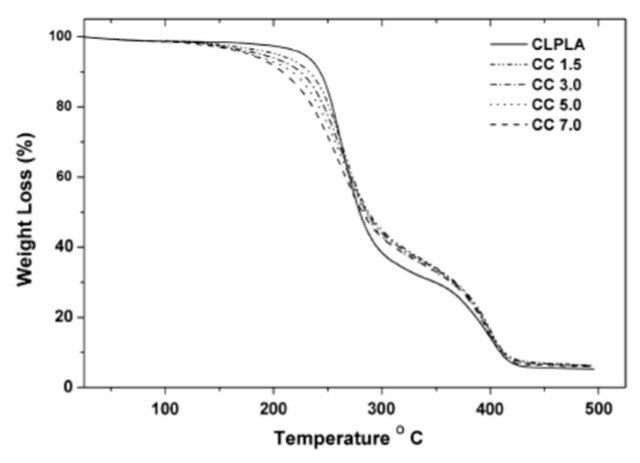
(**a**) TGA thermogram of CL–PLA/CC nanocomposites [45], *Reproduced with permission from Kumar V, Composites: Part B; published by Elsevier, 2014*.

**Figure 12 polymers-12-00732-f012:**
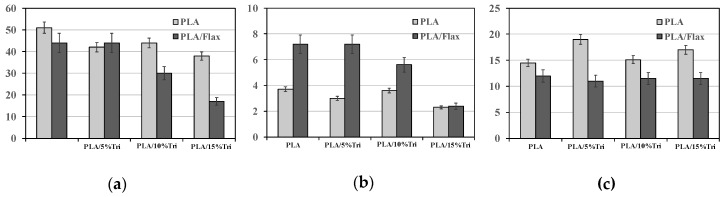
Mechanical properties: (**a**) tensile stress, (**b**) tensile modulus, and (**c**) unnotched Charpy impact strength of neat PLA and PLA/flax, without and with triacetin [50], *Reproduced with permission from Oksman K, Compos. Sci. Technol.; published by Elsevier, 2003*.

**Figure 13 polymers-12-00732-f013:**
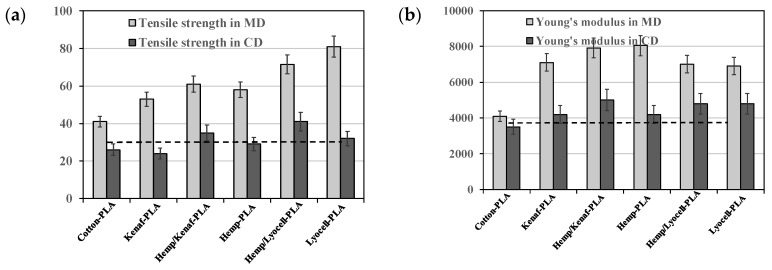
Mechanical properties: (**a**) Tensile strength, (**b**)Young’s modulus, (**c**) elongation at break, and (**d**) Charpy impact strength of the composites and neat PLA (MD—machine direction; CD—cross direction; standard deviation are shown as error bars; the letter in the figures a, b, c, d, and e means that there are significant differences between the mean values measured in MD) [67], *Reproduced with permission from Graupner N, Composites: Part A; published by Elsevier, 2009*.

**Figure 14 polymers-12-00732-f014:**
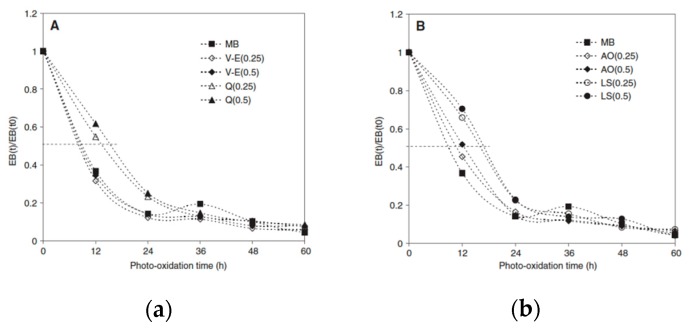
Dimensionless elongation at break, EB_(t)_/EB_(t0)_, of the starch-based biodegradable film (Mater-Bi^®^), additivated with (**a**) vitamin E (V-E) and quercetin (Q) and (**b**) synthetic antioxidant (AO) and light stabilizer (LS) [79], *Reproduced with permission from Dintcheva NTz, J. Polym. Eng.; published by De Gruyter, 2014*.

**Figure 15 polymers-12-00732-f015:**
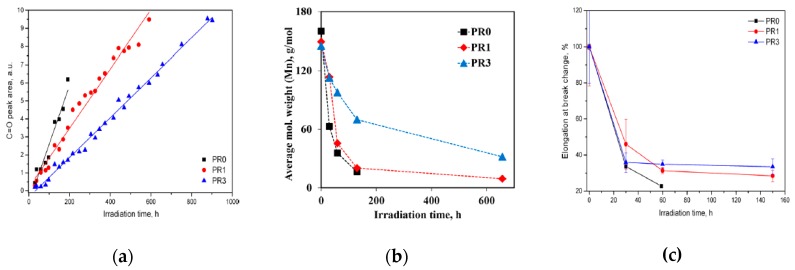
(**a**) Carbonyl functionalities, (**b**) molecular weight variation, and (**c**) elongation at break of neat PLA (PR0) and PLA containing 1 and 3 wt.% of R (PR1 and PR3) as a function of the irradiation time [82], *Reproduced with permission from Agustin-Salazar S, ACS Sustainable Chem. Eng.; published by ACS Publications, 2014*.

**Figure 16 polymers-12-00732-f016:**
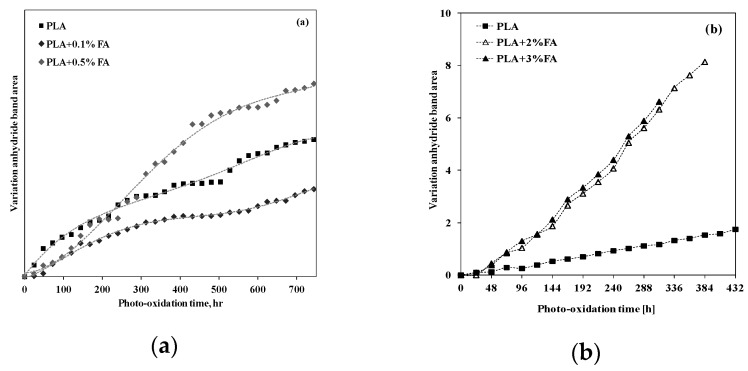
Variation of anhydride band area for neat PLA and PLA containing (**a**) 0.5 and 1 wt.% [93], *Reproduced with permission from Dintcheva NTz, Polym. Degrad. Stab.; published by Elsevier, 2017,* and (**b**) 2 and 3 wt.% of ferulic acid (FA) [26], *Reproduced with permission from Dintcheva NTz, Polym. Degrad. Stab.; published by Elsevier, 2018*.

**Table 1 polymers-12-00732-t001:** E′ value of nanocomposites of PLA at different temperature ranges. [37], Reproduced with permission from Fukushima K, Mater. Sci. Eng. C; published by Elsevier, 2009.

Sample	E’ at 30 °C (MPa)	E’ at 50 °C (MPa)	E’ at 60 °C (MPa)	E’ at 80 °C (MPa)
PLA	3058	2848	1544	5
PLA + 5% CLO30b	3578	3372	2761	12
PLA + 5% NAN804	3109	2902	2462	6
PLA + 5% SEPS9	3815	3579	2246	10

**Table 2 polymers-12-00732-t002:** E’ value of nanocomposites of PCL at different temperature ranges. [37], Reproduced with permission from Fukushima K, Mater. Sci. Eng. C; published by Elsevier, 2009.

Sample	E’ at −125 °C (MPa)	E’ at −75 °C (MPa)	E’ at −25 °C (MPa)	E’ at 30 °C (MPa)
PCL	4715	3451	891	503
PCL + 5% CLO30b	5071	3882	1312	762
PCL + 5% NAN804	4864	3663	1075	599
PCL + 5% SEPS9	4767	3602	1207	751

**Table 3 polymers-12-00732-t003:** Mechanical properties of virgin polylactic acid (PLA) and organo-modified layered montmorillonite (OMMT)—containing PLA [39].

Labels	Flexural Strength (MPa)	Flexural Modulus (GPa)	Tensile Strength (MPa)	Young’s Modulus (GPa)	Impact Strength (J/m^2^)
PLA	33.5 (2.4)	2.54 (0.35)	29.0 (1.8)	1.21 (0.05)	3.4 (0.4)
0.5/PLA	57.2 (3.8)	3.08 (0.40)	43.9 (2.2)	1.29 (0.09)	10.9 (2.2)
1/PLA	42.2 (4.5)	3.59 (0.18)	37.3 (1.0)	1.36 (0.05)	7.3 (0.6)
1.5/PLA	38.9 (4.0)	4.72 (0.17)	35.2 (2.1)	2.13 (0.18)	6.0 (0.7)
2/PLA	38.1 (2.0)	3.49 (0.22)	29.1 (0.9)	1.54 (0.20)	4.8 (0.7)

*Note:* Values in the parentheses represent the standard deviations of replicates.

**Table 4 polymers-12-00732-t004:** Chemical composition of OMMT (Cloisite 15A), determined by energy dispersive X-ray analysis (EDX) [28], *Reproduced with permission from Dintcheva NTz, Polym. Degrad. Stab.; published by Elsevier, 2009*.

**Element**		**Wt %**	**At %**	**K-Ratio**	**Z**	**A**	**F**
C	K	26.97	52.20	0.0566	1.1058	0.1896	1.0001
O	K	11-45	16.64	0.0343	1.0845	0.2756	1.0004
MgK		1.12	1.07	0.0080	1.0354	0.6855	1.0053
AlK		7.96	6.85	0.0633	1.0101	0.7826	1.0068
SiK		21.10	17.54	0.1795	1.0453	0.8099	1.0001
AuM		24.61	2.90	0.1721	0.7099	0.9850	1.0000
FeK		6.70	2.79	0.0609	0.9169	0-9869	1.0052
Total:		100.00	100.00				
**Element**		**Net Inte.**	**Bkgd Inte.**	**Inte. Error**	**P/B**		
C	K	6.36	0.91	2.76	7.00		
O	K	3.06	1.74	5.13	1.76		
MgK		4.42	5.94	5.61	0.74		
AlK		36.50	5.91	1.17	6.17		
SiK		107.25	6.44	0.63	16.65		
AuM		66.12	6.63	0.83	9.97		
FeK		7.53	3.41	3.09	2.21		
Total:							

**Table 5 polymers-12-00732-t005:** Values of the tensile strength (TS), Young’s modulus (YM), and elongation at break (EB) [43], *Reproduced with permission from Hajibeygi M, Polym. Adv. Technol.; published by Wiley, 2019*.

	Tensile Strength, MPa	Young Modulus, GPa	Elongation at Break, %
PLA	44.89 ± 3.21	2.41 ± 0.11	2.46 ± 0.11
PP8	64.17 ± 2.26	2.92 ± 0.18	2.45 ± 0.22
PH8	52.30 ± 1.89	2.63 ± 0.23	2.79 ± 0.29
PP2H6	88.07 ± 2.80	3.71 ± 0.13	2.89 ± 0.10
PP6H2	95.58 ± 3.54	3.42 ± 0.14	3.21 ± 0.18
PP4H4	88.17 ± 3.80	3.24 ± 0.36	3.00 ± 0.23

**Table 6 polymers-12-00732-t006:** TGA data of CL–PLA/CC nanocomposites [45], *Reproduced with permission from Kumar V, Composites: Part B; published by Elsevier, 2014*.

Sample	T_10_	T_50_
CC 1.5	238	282
CC 3.0	231	281
CC5.0	224	281
CC 7.0	213	279
CL PLA	246	278

**Table 7 polymers-12-00732-t007:** Basic chemical composition of some common natural fibers [71], *Reproduced with permission from Faruk O, Progr. Polym. Sci.; published by Elsevier, 2012*.

Fibers	Cellulose, wt.%	Hemicellulose, wt.%	Lignin, wt.%	Waxes, wt.%
*Flax*	71	18–20	2.2	1.5
*Hemp*	68	15	10	0.8
*Jute*	61–71	14–20	12–13	0.5
*Kenaf*	72	20	9	-
*Sisal*	65	12	10	2
*Bamboo*	26–43	30	21–31	-
*Oil palm*	65	-	29	-
*Rice husk*	35–45	19–25	20	14–17

**Table 8 polymers-12-00732-t008:** Basic chemical structures of most common natural antioxidants [23], *Reproduced with permission from Dintcheva NTz, ACS Sustainable Chem. Eng.; published by ACS Publications, 2019*.

Natural Antioxidants (nAO)	Chemical Formula	Natural Antioxidants (nAO)	Chemical Formula
*Hydroxycinnamic acids*	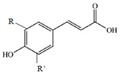	*Resveratrol*	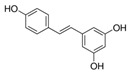
*Hydroxybenzoic acids*	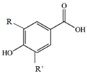	*Catechin*	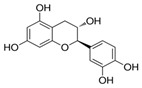
*Flavene backbone*	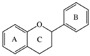	*Vitamin E*	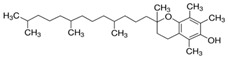
*Quercetin*	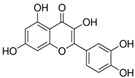	*β-carotene*	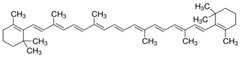

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
