# Peer review of "Natural Compounds as Sustainable Additives for Biopolymers"

_polymers, 2020, doi:10.3390/polym12040732_

Round 1

Reviewer 1 Report

In this review article, the authors discussed the application of three types of naturally occurring compounds as the additives of biopolymers. The topic is interesting and is important for the researchers working in the area of polymers and composites. However, the manuscript needs to be major revised before it can be accepted for publication.
1. What is the reason of dividing the additives to three groups of Nanofillers, Nanofibers, and anti-oxidant molecules? Other types of commonly used additives such as cellulose nanocrystals were not included in the review.
2. The authors mostly described the addition of additives into polymers such as PLA and PLLA. But other types of biopolymers were discussed very shortly.
3. There are some mistakes about the language and format. For example, on Page 9, “hydrohyapatite (HA)” should be “hydroxyapatite”? Some abbreviations have been defined several times. The number of the figures and tables are not in the right order. After Figure 12, it comes to Figure 1.
4. The applications of the composites are not well discussed.
5. What are the major limitations for the modification of the biopolymers using these natural additives?
6. As a review article, a future perspective should be provided.

Author Response

First of all, we would thank to the reviewer for his/her valuable comments and suggestions. Our answers to reviewer’s comments are reported following:

  1. The reason of dividing the natural compounds as fillers, fibers and anti-oxidants, is related to main benefits which can be obtained by adding of these compounds to biopolymers, please see paragraph 1.3. There is cited the main classes of additives which can improve the biopolymers macroscopical properties and performance, such as rigidity, mechanical and thermo-mechanical resistance, barrier properties and resistance to oxidation, i.e. durability. However, in the original version of review, as suitable spherical particles for biopolymers, have been considered different kinds of alumino-silicates and calcium carbonate. In the revised version of manuscript, the nanocrystalline cellulose has been considered and a new paragraph has been added; please, see new section 2.4. Thank you of reviewer for this suggestion, which is valuable to improve the quality of our review.
  1. The focus of this review is related to the “natural compounds” as additives for biopolymers (not on the polymer matrix kind), and for this reason, we have considered mainly PLA, PLLA and PLA-based blends (for example, PLA/PCL) in order to emphasize the effects of different naturally occurring compounds. Additionally, other biopolymers such as PHA, PBS, etc., are discussed shortly, also considering their limited current industrial applications.
  2. Thank you for highlighting these mistakes, the wrong word “hydrohyapatite” has been replaced with “hydroxyapatite” and the figures has been placed in correct order. We are sorry for these mistakes.
  3. The aim of this review is the presentation of main kind of natural compounds (fillers, fibers and anti-oxidants), following the scheme: chemical structure, usefulness and applications, preparation of biopolymers-based compounds, using examples from literature. The main applications fields have been cited (for example, packaging, biomedical and pharmaceutical sectors), but there is not discussed largely about the applications also because the applications fields of new biopolymer-based materials must be consolidated and established, as reported in the manuscript.
  4. Concerning the major limitations of biopolymers using natural additives, in the manuscript are cited some drawbacks and disadvantages using naturally occurring compounds, for example, the introduction of hydroxyapatite to semi-crystalline PLA leads to decrease of crystalline degree and these nanocomposites show reduced durability, additionally, the large amount of anti-oxidants, such as vanillic acid, resveratrol, vitamin-e, induces a pro-oxidant actions, which leads to a significant reduction of durability, rather than stabilization action and improved durability. These example has been discussed shortly in the text.
  5. A short paragraph related to future perspective in using naturally occurring compounds has been added in section “Conclusion remarks and future perspectives”. Thank you for this valuable suggestion.

Reviewer 2 Report

The paper entitled "Natural Compounds as Sustainable Additives for Biopolymers" by Nadka Tzankova Dintcheva and co. is a short review in which the authors presented some aspects concerning few natural compounds.

Some changes may be done in order to improve, from the scientific point of view, the content of the paper:

1) The Introduction contains unnecessary data as figure 1 (This figure is not necessary because the text is enough for demonstrating the growing popularity of the subject).  

2) Some aspects as Applications of natural compounds as sustainable Additives for biopolymers must be introduced in one separated paragraph (maybe before Conclusions). Data concerning the main applications highlighting the advantages of using natural compounds must be introduced in the paper, by using examples from literature.

3)  The word "vantages" must be replaced with "advantages" for a better understanding of the scope (in Abstract and Conclusions).

Author Response

First of all, we would thank to the reviewer for his/her positive comments. Our answers to reviewer’s comments are reported following:

1) In the text is reported only the number of scientific papers containing the keyword “Natural Compounds”, i.e. 14.040 papers in 2019, based on Scopus data on January 2020, while the Figure 1 shows the trend of published paper, having same keywords, since 1980 until today. We are sorry, but the information delivered by text and Figure 1 are different, particularly, the text gives information about the current status and the figure gives a clear idea about the growing interest towards this fields. The authors believe that the Figure 1 is useful in highlighting the exponential trend toward this research topic.

2) A separated paragraph about the application has not been introduced because the usefulness of each cited kind of natural compounds, i.e. fillers, fibers and anti-oxidant, has been commented in the text. The aim of this review is the presentation of main kind of natural compounds, following the scheme: chemical structure, usefulness and applications, preparation of biopolymers-based compounds, using examples from literature.

3) Thank you for this observation, the wrong “vantages” has been replaced with “advantages”.

Round 2

Reviewer 1 Report

The authors have revised the manuscript properly. There are still some grammatical and typomistakes, abbreviation problems, which need to be carefully revised before publication.

Author Response

Thank you very much for your positive evaluation. We check languages and some grammar mistakes/errores have been corrected. Please see revised manuscript.

We hope that in this version the manuscript can be considered suitable for publication on Polymers.

Reviewer 2 Report

1) I agree that  "the usefulness of each cited kind of natural compounds, i.e. fillers, fibers and anti-oxidant, has been commented in the text", but in some paragraphs on a general manner like "industrial application", "biopolymers application fields",' "some packaging applications", "outdoor applications". 

For example, in paragraph "2.2. Natural hydroxyapatite", was mentioned that "the adding of hydroxyapatite (nHA), has beneficial effects on the thermo-mechanical, mechanical and resorption behaviour of the PLLA-based nanocomposites" for"biomedical applications, such as porous biodegradable scaffolds for bone repair". Please do this in all paragraphs. 

2)The wrong “vantages” has not been replaced with “advantages” in all the paper as the authors claimed (see pag 18 line 667)

Author Response

Thank you for your suggestions.

As requested, we add in the text for each kind of naturally occuirng fillers, fibers and anti-oxidants, most important application fields of biopolymer-based materials containing these additives.

Additionally, some grammar erros and mistakes have been corrected.

Please see revised version of manuscipt.

We hope that in this version the manuscript can be considered suitable for publication on Polymers.